How glyphosate and its associated acidity affect early development in zebrafish (Danio rerio)

Schweizer Mona 1 mona.schweizer@gmx.de
Brilisauer Klaus 2
Triebskorn Rita 1 3
Forchhammer Karl 2
http://orcid.org/0000-0002-5160-3499 Köhler Heinz-R. 1
1 Institute of Evolution and Ecology, Animal Physiological Ecology, Eberhard-Karls-Universität Tübingen , Tübingen , Germany
2 Microbiology, Organismic Interactions, Eberhard-Karls-Universität Tübingen , Tübingen , Germany
3 Steinbeis Transfer-Center for Ecotoxicology and Ecophysiology , Rottenburg am Neckar , Germany
Anderson Todd
Electronic publication date: 2019 Jun 19
Publication date: 2019
Volume: 7
Electronic Location ID: e7094
Received 2019 Feb 25; Accepted 2019 May 7
Copyright: © 2019 Schweizer et al.
Copyright year: 2019
Copyright holder: Schweizer et al.
License: This is an open access article distributed under the terms of the Creative Commons Attribution License, which permits unrestricted use, distribution, reproduction and adaptation in any medium and for any purpose provided that it is properly attributed. For attribution, the original author(s), title, publication source (PeerJ) and either DOI or URL of the article must be cited.
License URL: https://creativecommons.org/licenses/by/4.0/

Keywords: Fish embryo test, Glyphosate effects, pH, Developmental toxicity, 7dSh, Danio rerio

Funding: The authors received no funding for this work.

==============================
Background

Glyphosate is among the most extensively used pesticides worldwide. Following the ongoing highly controversial debate on this compound, its potential impact on non-target organisms is a fundamental scientific issue. In its pure compound form, glyphosate is known for its acidic properties.

Methods

We exposed zebrafish (Danio rerio) embryos to concentrations between 10 μM and 10 mM glyphosate in an unbuffered aqueous medium, as well as at pH 7, for 96 hours post fertilization (hpf). Furthermore, we investigated the effects of aqueous media in the range of pH 3 to 8, in comparison with 1 mM glyphosate treatment at the respective pH levels. Additionally, we exposed zebrafish to 7-deoxy-sedoheptulose (7dSh), another substance that interferes with the shikimate pathway by a mechanism analogous to that of glyphosate, at a concentration of one mM. The observed endpoints included mortality, the hatching rate, developmental delays at 24 hpf, the heart rate at 48 hpf and the malformation rate at 96 hpf. LC10/50, EC10 and, if reasonable, EC50 values were determined for unbuffered glyphosate.

Results

The results revealed high mortalities in all treatments associated with low pH, including high concentrations of unbuffered glyphosate (>500 μM), low pH controls and glyphosate treatments with pH < 3.4. Sublethal endpoints like developmental delays and malformations occurred mainly at higher concentrations of unbuffered glyphosate. In contrast, effects on the hatching rate became particularly prominent in treatments at pH 7, showing that glyphosate significantly accelerates hatching compared with the control and 7dSh, even at the lowest tested concentration. Glyphosate also affected the heart rate, resulting in alterations both at pH 7 and, even more pronounced, in the unbuffered system. In higher concentrations, glyphosate tended to accelerate the heart rate in zebrafish embryos, again, when not masked by the decelerating influence of its low pH. At pH > 4, no mortality occurred, neither in the control nor in glyphosate treatments. At 1 mM, 7dSh did not induce any mortality, developmental delays or malformations; only slightly accelerated hatching and a decelerated heart rate were observed. Our results demonstrate that lethal impacts in zebrafish embryos can be attributed mainly to low pH, but we could also show a pH-independent effect of glyphosate on the development of zebrafish embryos on a sublethal level.

Introduction

Currently, glyphosate is the most extensively used herbicide worldwide (The European Commission, 2007; U.S. EPA, 2011), and its approval for application within European Union borders was extended by the European Commission (The European Commission, 2017) for another 5 years in December 2017. Glyphosate was first used successfully in the 1970s as the active compound in the formulation Roundup® (Monsanto, St. Louis, MO, USA) and is nowadays a component in numerous herbicides from various producers. In the U.S., glyphosate tops the list of applied agricultural herbicides and ranks second in home and garden applications (Battaglin et al., 2014). However, in the last couple of years, there have been very controversial debates concerning the effects of glyphosate-based herbicides on non-target organisms (Folmar, Sanders & Julin, 1979; Relyea, 2005; Benamú, Schneider & Sánchez, 2010; Balbuena et al., 2015; Helmer et al., 2015; Motta, Raymann & Moran, 2018), whole ecosystems (Vera et al., 2010) and even on the health of human beings as the International Agency for Research on Cancer (IARC) rated glyphosate as probably carcinogenic in 2015 (International Agency for Research on Cancer (IARC), 2017).

Glyphosate is a non-selective, broad-spectrum herbicide (Monsanto, 2005) that inhibits aromatic acid synthesis in plants by interfering with the shikimate pathway (Steinrücken & Amrhein, 1980; Bentley & Haslam, 1990; Schönbrunn et al., 2001). Since the shikimate pathway is specific to plants, fungi, bacteria and some protists but is absent in animals, including humans (Maeda & Dudareva, 2012), glyphosate was regarded an ideal and safe application option. However, recent studies suggest the metabolization of glyphosate by humans leading to adverse effects on human health, already caused by chronic low-dose glyphosate ingestion via residues in food and water (Swanson, Hoy & Seneff, 2016). Those potential effects include sub-clinical metabolic acidosis induced by shifts in the human gut microbiome, mitochondrial dysfunction, endocrine disruption, DNA damage and the inhibition of the cytochrome P450 enzyme system (Samsel & Seneff, 2013; Swanson et al., 2014; Swanson, Hoy & Seneff, 2016). Concerning application in the field, glyphosate is advertised as a herbicide with poor rainfastness (Baylis, 2000), is known to bind to soil particles and is degraded by microbes (Rueppel et al., 1977; von Wirén-Lehr et al., 1997; Forlani et al., 1999). Although Giesy, Dobson & Solomon (2000) stated that less than 1% of glyphosate is washed from the leaves of plants by rainfall directly after application, spray drift and runoff may occur that contribute to the entry of glyphosate into non-target environments, particularly aquatic systems (Giesy, Dobson & Solomon, 2000). Furthermore, incidences of for example, incorrect disposal of packages and expired herbicide remnants or careless cleansing of application instruments might enhance the amount of glyphosate entering the environment.

Since glyphosate is applied as formulation, additives like polyoxyethylene tallow amine (POEA), a surfactant that facilitates glyphosate uptake through the plant cuticula, are released into ecological systems inducing adverse effects on non-target organisms at least as severe as glyphosate itself (Perkins, Boermans & Stephenson, 2000; Tsui & Chu, 2003; Brausch, Beall & Smith, 2007; Bringolf et al., 2007; Moore et al., 2012; Defarge, Spiroux De Vendômois & Séralini, 2018). However, glyphosate is not monitored routinely (Uren Webster et al., 2014), and numerous studies show that it is transported from fields (Struger et al., 2008), national parks (Battaglin et al., 2009) and urban areas (Botta et al., 2009) into surface waters and even groundwater. In the environment, glyphosate is degraded to CO2 and aminomethylphosphonic acid, which, in turn, is degraded to inorganic phosphate, ammonium and CO2. Glyphosate’s half-life depends on the surrounding matrix and is estimated between 4.2 and 14 days for fresh water (Giesy, Dobson & Solomon, 2000; Vera et al., 2010), mainly below 60 days in agricultural soils (Giesy, Dobson & Solomon, 2000; Grunewald et al., 2001) and between 47 and 315 days in seawater (Mercurio et al., 2014). Depending on time and location, Struger et al. (2008) measured a mean maximum glyphosate concentration in surface waters between 10 and 20 μg/L, peaking at up to 40 μg/L. Botta et al. (2009) could detect 75 to 90 μg/L in storm sewers after rainfall, and Battaglin et al. (2014) even found top concentrations up to 301 μg/L in lake/wetland samples. Although those findings range below the U.S. EPA’s (2009) maximum contamination level of 700 μg/L or the Canadian long- and short-term freshwater aquatic life standards of 800 and 27 000 μg/L, respectively (Canadian Council of Ministers of the Environment, 2012), adverse effects in non-target organisms in the environment have been demonstrated, as already indicated. Therefore, it is of major importance to further elucidate the mechanisms by which glyphosate affects non-target organisms and to assess its risk more realistically. In the long run, it could be conceivable to further establish ecologically sound alternatives, including bioherbicides and synthetic herbicides on the basis of natural phytotoxins (Dayan & Duke, 2014), that can potentially substitute for glyphosate.

Since this study is regarded as fundamental research, the zebrafish Danio rerio, as a prominent and easy-to-handle standard organism in the field of ecotoxicology, was used, and the glyphosate concentrations chosen were rather high (one μM (1.69 mg/L)—10 mM (1.69 g/L)) and less environmentally relevant, as glyphosate concentrations measured in surface waters lie mainly within a range of μg/L (Struger et al., 2008; Botta et al., 2009; Battaglin et al., 2014). Glyphosate, in its pure compound form, is known to be a highly polar molecule, a polyprotic acid with pKa values of <2, 2.3 and 5.6 (Bromilow et al., 1993; Wade et al., 1993; Borggaard & Gimsing, 2008). Consequently, it lowers the pH of the medium when in solution. As an ionisable substance, glyphosate exerts a higher toxicity in low pH environments, where a higher proportion of the molecules are present in a neutral state, in which an uptake through biomembranes is facilitated, compared to glyphosate ions (Erickson et al., 2006; Saparov, Antonenko & Pohl, 2006; Ehrl et al., 2018). Thus, it might be expected that glyphosate toxicity is enhanced in systems dealing with acidification already, whereas in systems characterized by more alkaline surroundings, the toxicity might be mitigated. Although glyphosate has been studied regarding its impact on non-target organisms, including adverse effects on early stages in zebrafish (Sulukan et al., 2017; Zhang et al., 2017; Fiorino et al., 2018), the influence of pH on toxicity results have not been considered so far.

The aim of our study was to differentiate the effects of glyphosate-induced acidification of the medium and those exerted by the compound itself (independent of low pH) on embryonic and early larval development of D. rerio. This study should provide lethal and effect concentration data (LC/EC10 and LC/EC50) for a more profound risk assessment. Besides mortality, additional sublethal endpoints, including hatching, the heart rate, developmental delays and malformations, were observed to account for subtler impacts on embryonic health. For that purpose, glyphosate was tested at concentrations between 10 μM and 10 mM in an unbuffered medium and in an aqueous medium adjusted to pH 7. In addition, the critical range of pH values was determined and tested with and without one mM glyphosate. To investigate whether possible effects are specific to glyphosate or to general inhibition of the shikimate pathway, we tested the newly described shikimate pathway inhibitor, 7-deoxy-sedoheptulose (7dSh). This substance is produced by cyanobacteria and was used at a concentration of one mM, which is more than one order of magnitude above the effective concentration required to inhibit the shikimate pathway (Brilisauer et al., 2019).

Material and Methods

Glyphosate

Glyphosate (N-(phosphonomethyl)glycine, 96% pure substance, molecular weight: 169.07 g/mol, CAS: 1071-83-6; Sigma-Aldrich, Merck KGaA, Darmstadt, Germany) was used to prepare the test solutions. A stock solution with a concentration of 25 mM was prepared as follows: glyphosate was diluted in reconstituted water (0.23 g KCl, 2.59 g NaHCO3, 4.93 g MgS4O 7 H2O and 11.76 g CaCl2 2 H2O were dissolved separately in one L double-distilled water, then 25 mL of each stock solution was added to 900 mL double-distilled water). The stock solution was then diluted to the following test concentrations: 10, 50, 100, 250, 500, 750 μM, one and 10 mM glyphosate (respective specifications for mg/L and pH are given in Table 1). All those concentrations were tested unbuffered and at pH 7. For pH adjustments, 1M HCl and NaOH solutions were used as recommended in the Organisation for Economic Co-operation and Development (OECD) 236 (2013) guideline. For investigations of the influence of pH, 1 mM glyphosate was tested at different pH values ranging between pH 3 and 8 vs. the respective pH controls. Due to preliminary results from the broad-scale pH testing, particular attention was paid to the range between pH 3 and 4. Measurements of pH were conducted with a pH meter (SevenCompactDuo; Mettler Toledo, Gießen, Germany) directly prior to the exposure.

Table 1 Tested glyphosate concentrations including their corresponding values in mg/L and resulting pH in solution.

Molarity	mg/L	pH	
10 μM	1.69	8.3	
50 μM	8.45	6.9	
100 μM	16.91	5.9	
250 μM	42.27	4.5	
500 μM	84.54	3.6	
750 μM	126.80	3.4	
1 mM	169.07	3.2	
10 mM	1,690.7	2.4	

7-deoxy-sedoheptulose (7-deoxy-D-altro-2-heptulose, 7dSh)

The compound 7dSh was obtained by chemoenzymatic synthesis and purified as described in Brilisauer et al. (2019). This 7dSh was tested at a one mM concentration alongside the unbuffered glyphosate treatments in an unbuffered system. Thus, 7dSh was tested as a neutral solution, since it has no influence on pH.

Maintenance of zebrafish and test procedure

The embryos used in this study stem from our own breeding stock of the D. rerio (Hamilton, 1822) Westaquarium strain established in the Animal Physiological Ecology group, Tübingen University. Adult zebrafish were kept in 90 L aquaria filled with a 1:1 mixture of purified water and filtered tap water (AE-2L water filter with an ABL-0240-29 activated carbon filter, 0.3 μm; Reiser, Seligenstadt, Germany) at 26 ± 1 °C and an oxygen saturation of 100% ± 5%. Conductivity ranged from 260 to 350 μS/cm, nitrite and nitrate concentrations from 0.025 to 0.1 mg/L, one and five mg/L, respectively, and total water hardness from eight to 12 dH. Fish were subjected to an artificial 12:12 h day/night cycle and fed three times daily with flake food (TetraMin®; Tetra GmbH, Melle, Germany) supplemented with frozen black mosquito larvae and glass worms (Poseidon Aquakultur Freeze, Ruppichteroth, Germany) prior to spawning to ensure sufficient dietary protein.

The day before the test, pre-exposure and test Petri dishes (90 and 30 mm in diameter) were filled with the respective solutions and stored at 26 ± 1 °C overnight to saturate the glass (the same was done with the Schott flask used for the stock solution, beforehand). On the morning of the test, Petri dishes were emptied and refilled with 70 mL (pre-exposure) and three mL (test Petri dishes) solution. For spawning, Plexiglas® boxes 20 × 20 × 6 cm in size and covered with a mesh grid to keep zebrafish from feeding on their own eggs were used as breeding boxes. They were topped with artificial sea grass acting as an optical spawning stimulus and were placed into the fish tanks the evening before the start of the test. Zebrafish spawn at sunrise; therefore, spawning in the laboratory starts with the onset of light the next morning. Eggs were collected with a sieve, rinsed with tepid tap water, transferred into pre-exposure Petri dishes and incubated for 2 h at 26 ± 1 °C. Following the pre-exposure, eggs for the test were chosen with regard to their age and developmental stage (0 hours post fertilization (hpf) ≙ 8 a.m.), placed into the small 30 mm Petri dishes and stored in a heated cabinet at 26 ± 1 °C. A total of 32 individuals were used per treatment, that is, four per Petri dish and eight replicates each. Embryos were checked every 12 to 24 h. Endpoints investigated under a stereo microscope (Stemi 2000-C; Zeiss, Oberkochen, Germany) included mortality, developmental delays at 24 hpf, heart rate at 48 hpf, hatching success from 60 to 96 hpf and malformations at 96 hpf (see Table 2). Except for mortality, analysis of all endpoints, including hatching success, was based on living embryos/larvae at the respective time point of evaluation. Heart rates were determined from two out of four individuals per Petri dish for 20 s, and values were extrapolated to 1 min. Coagulated eggs, dead larvae and empty egg shells were removed from the Petri dishes to avoid depletion of oxygen due to biological degradation processes. The embryo test was run three times and conducted according to Organisation for Economic Co-operation and Development (OECD) 236 (2013). The compound 3,4-dichloraniline (98%, CAS: 95-76-1; Sigma-Aldrich, Merck KGaA, Darmstadt, Germany) at a concentration of four mg/L served as a positive control and reconstituted water, as a negative control.

Table 2 Overview of observed lethal and sublethal endpoints at respective time points.

Endpoint	12 hpf	24 hpf	48 hpf	60 hpf	72 hpf	96 hpf	
Mortality	✓	✓	✓	✓	✓	✓	
Developmental delays		✓					
 No somites		✓					
 Non-detachment of the tail		✓					
 No development of the eyes		✓					
Heart rate			✓				
Hatching success				✓	✓	✓	
Malformations						✓	
 Oedema						✓	
 Eye/brain defects						✓	
 Deformation of the spine						✓	
 Light pigmentation						✓	

According to the Directive 2010/63/EU of the European Parliament and the Council on the protection of animals for scientific purposes, D. rerio embryos and larvae that do not feed independently are not regarded as animals, thus regulations and permissions for animal testing do not apply. Nevertheless, all embryos in our tests were handled in the least stressful way possible and with the utmost care. After test termination embryos/larvae were euthanized with MS222.

Statistics

All statistical analyses were conducted in JMP® 11.2.0 (SAS Institute Inc., Cary, NC, USA). Mortality, hatching success and the malformation rate at 96 hpf, as well as developmental delays at 24 hpf, were analysed with a likelihood-ratio χ2 test, followed by Fisher’s exact test. Finally, the sequential Bonferroni-Holm method was applied accounting for multiple testing. A Cox regression was used to assess mortality and hatching success over time. For the analysis of heart rate, the data were averaged per Petri dish and checked for a normal distribution and homogeneity of variances. Subsequently an ANOVA with Tukey’s HSD or Dunnett’s test was conducted. If data did not meet the criteria for an ANOVA and transformation of the data did not lead to the desired result, a non-parametric Steel-Dwass test was conducted instead. Additionally, for assessing the pH range in which pH control and glyphosate treatments differed in heart rate across the whole span of tested pH, non-linear regression analysis, including calculation of 95% confidence intervals (TableCurve 2D v5.01; SYSTAT Software Inc., San Jose, CA, USA), was applied. Non-linear regression analysis by TableCurve was also used for determining LC10/EC10 and LC50/EC50 values of endpoints in unbuffered glyphosate treatments.

Results

After 96 hpf, mortality and hatching success were 0% and above 80%, respectively, in control embryos. The 3,4-dichloraniline positive control induced high mortalities, with rates consistently above 80% after 96 hpf. Thus, the validity criteria according to Organisation for Economic Co-operation and Development (OECD) 236 (2013), including sensitivity of zebrafish, were met.

Unbuffered glyphosate

At the two highest concentrations tested (one and 10 mM), it was already difficult to select well-developed eggs after the 2 h pre-exposure period. The yolk sac, which usually has a regular spherical shape, was found to be asymmetric and partly oval, and the chorion fluid, which is naturally clear, was murky in some cases and contained indefinable streaks (Fig. 1). As early as 12 hpf, all individuals, without exception, in the 10 mM treatment died (Fig. 2A). Mortality in the 1 mM exposure experiment was beyond 85% at 12 hpf and reached 100% within the first 24 h. Within the 750 μM glyphosate treatment, only six out of a total of 96 individuals survived until the end of the test at 96 hpf, whereas concentrations of 250 μM and below resulted in negligible or no mortality (≤3.125%). Regarding mortality at 96 hpf, all treatments ≥500 μM were highly significantly different from the control (likelihood ratio χ2, p < 0.001). Lethal concentrations were calculated to be 385 μM (LC10) and 582 μM (LC50) at 96 hpf.

Figure 1 Danio rerio embryos at 3.3 hpf (“high stage,” blastula period, according to Kimmel et al. (1995)).

(A) Control embryo; (B–D) exemplary embryos with irregularly growing cellular “hood” or misshapen yolk sacs from 1 mM glyphosate concentration; as seen at five times magnification under a stereo microscope (Stemi 2000-C; Zeiss, Oberkochen, Germany).

Figure 2 Mortality, heart rate and hatching success in percentage of unbuffered and pH 7 treatment.

(A) Mortality after 96 hpf (likelihood ratio χ2, Fisher’s exact test, Bonferroni-Holm, p < α), (B) heart rate at 48 hpf relative to the negative control (Steel-Dwass, p < 0.01), (C) hatching rate over time in unbuffered treatments (Cox regression, p < 0.05), (D) hatching rate in pH 7 treatments over time (Cox regression, p < 0.001); shaded bars mark treatments with n < 5 that show tendencies but are not included in the statistical analyses.

Heart rates showed a concentration-dependent relationship, decreasing with increasing glyphosate concentration (Fig. 2B). The mean heart rate was 149 beats per minute (bpm) for the control and between 130 and 140 bpm for low (10, 50 μM), 120 and 130 bpm for medium (100, 250 μM) and 110 and 120 bpm for the higher (500, 750 μM) concentrations. Thus, differences between the control and the 750 μM concentration ranged between 30 and 40 bpm. The treatments with the highest concentrations of glyphosate (one mM, 10 mM) could not be evaluated due to 100% mortality at that time point. Only two individuals out of those exposed to one mM glyphosate survived until 60 hpf and seemed to continue the observed relationship between glyphosate and heart rate by showing even lower rates (93 and 96 bpm). As single individuals, they were not included in the statistical analysis. All remaining treatments were significantly different from the control (ANOVA with Tukey’s HSD, p < 0.001) and the relationship between glyphosate concentration and heart rate could be described by linear regression analysis (R2 = 0.546074, p < 0.001). The EC10 was 43 μM.

Concerning the hatching rate, we observed a clear division between a cluster of treatments that comprised the control treatment and lower concentrations of glyphosate (10, 50, 100 μM) and another treatment cluster comprising higher concentrations (250, 500, 750 μM) (Fig. 2C). Embryos exposed to lower concentrations hatched in 98–100% of cases, whereas hatching success in the experiments with 250 and 500 μM glyphosate was approximately 40%. All glyphosate treatments showed significant differences compared with the control (Cox regression, p < 0.05). EC10 and EC50 levels at 96 hpf were 155 and 224 μM, respectively.

There were no developmental delays at 24 hpf for glyphosate concentrations between 10 and 100 μM, whereas in treatments with 250 to 750 μM, rates varied from 15% to 25% (data summarized in Table 3). The EC10 for this endpoint was 126 μM. Results for all concentrations ≥250 μM were highly significant (likelihood ratio χ2, p < 0.001) compared with the control. A direct concentration dependency could not be observed. Rather, it seemed that a distinct concentration threshold had to be exceeded to induce those developmental delays and failures, which later approached the same level. Prevalent defects were a lack of tail detachment, sometimes combined with apically curved tails (Fig. 3F); a lack of somite formation and an impairment of eye development was not detected. Occasionally, embryos were fully developed but either the complete tail or just the posterior end of their tails remained attached to the yolk sac (Fig. 3E). Under normal conditions, movement begins after tail detachment. Yet, even the embryos in glyphosate treatments that lacked tail detachment, overall development had progressed to a point at which muscular contractions were already visible. But due to the undetached tails, embryos were unable to turn around and their movement was very limited. Additionally, some embryos had the posterior end of their tails detached but displayed severe spine deformations (Figs. 3G–3I). Those embryos could not move their tails in the same fluid manner as normally developed embryos could.

Table 3 Results for concentration-dependent glyphosate treatments, as well as for pH-dependent control and glyphosate treatments, as percentages.

	Mortality	Hatching	HR	D	M	
	96 hpf (%)	Over time	96 hpf (%)	Over time	48 hpf (bpm)	24 hpf (%)	96 hpf (%)	
Unbuffered	
 Neg. control	0	–	97.92	–	148.75	0	0.26	
 10 μM	1.04	n.s.	97.92	*	138.38***	0	2.36*	
 50 μM	0	n.s.	98.96	*	134.19***	0	5.47*	
 100 μM	0	n.s.	100	***	128.69***	0	4.69*	
 250 μM	3.13	n.s.	40.65*	***	125.56***	22.23*	13.57*	
 500 μM	28.13*	*	40.58*	***	118.63***	20.36*	16.06*	
 750 μM	93.64*	***	33.33*	n.a.	94.50***	19.04*	18.06*	
 1 mM	100*	***	n.a.	n.a.	n.a.	n.a.	n.a.	
 10 mM	100*	***	n.a.	n.a.	n.a.	n.a.	n.a.	
 LC10/EC10	385 μM	155 μM	43 μM	126 μM	179 μM	
 LC50/EC50	582 μM	224 μM	–	–	–	
 7dSh	0	n.s.	98.93	*	139.75***,†	0.35	0.27†	
Neutral (pH 7)	
 Neg. control	0	–	85.42	–	148.04	0	0.52	
 10 μM	1.04	n.s.	100*	***	144.25	0	1.04	
 50 μM	1.04	n.s.	100*	***	142.56	1.04	1.87	
 100 μM	0	n.s.	98.96*	***	136.57**	0	2.60	
 250 μM	0	n.s.	100*	***	130.31***	0	0.26	
 500 μM	0	n.s.	100*	***	129.19***	0	1.56	
 750 μM	1.04	n.s.	100*	***	133.50*	0	2.35	
 1 mM	0	n.s.	98.96*	***	145.94	0.69	6.56*	
 10 mM	0	n.s.	94.79*	***	149.38	0	7.29*	
pH range—control	
 Neg. control	0	–	94.69	–	160.38	0	0.26	
 pH 3	100*	***	n.a.	n.a.	n.a.	n.a.	n.a.	
 pH 3.1	100*	***	n.a.	n.a.	n.a.	n.a.	n.a.	
 pH 3.2	100*	***	n.a.	n.a.	n.a.	n.a.	n.a.	
 pH 3.3	84.03*	***	61.11*	***	140.19***	9.36*	4.17*	
 pH 3.4	51.04*	***	77.78*,†	***	141.38***	6.93*	3.51*	
 pH 3.5	8.33*	n.s.	15.77*,†	***,†	144.25***	8.32*	2.0*	
pH range—glyphosate	
 pH 3	100*	***	n.a.	n.a.	n.a.	n.a.	n.a.	
 pH 3.1	100*	***	n.a.	n.a.	n.a.	n.a.	n.a.	
 pH 3.2	100*	***	n.a.	n.a.	n.a.	n.a.	n.a.	
 pH 3.3	72.57*	***	83.33	***	136.46***	13.33*	0	
 pH 3.4	28.47*	***	60.61*,†	***	141.49***	4.54*	3.80*	
 pH 3.5	17.36*	n.s.	66.57*,†	***,†	143.50***	10.31*	2.04*	
Notes:

Asterisks (*) and bold indicate statistically significant differences from the negative control (Cox regression, ANOVA. *p < 0.05. **p < 0.01. ***p < 0.001. Likelihood ratio χ2, Fisher’s exact test, Bonferroni-Holm: *p < α).

Crosses (†) denote additional statistical significances between pH control and glyphosate within the same pH range or, in the case of 7dSh, differences from 1 mM glyphosate at pH 7. For unbuffered glyphosate concentrations, endpoint-related LC10/EC10 and LC50/EC50 values are given.

HR, heart rate; D, developmental delays; M, malformations; n.s, not significant; n.a., not available (no sufficient sample sizes for statistical analysis).

Figure 3 Danio rerio embryos at 24 hpf.

(A) Coagulated control embryo; (B) coagulated embryo in glyphosate (1 mM); (C) streaks within the chorion; (D) control embryo; (E) no detachment of the tail; (F) attached tail growing curved at the distal end; (G) dorsal dent; (H) irregular curved spine; (I) anomalously straight tail; as seen at five times magnification under a stereo microscope (Stemi 2000-C; Zeiss, Oberkochen, Germany).

Malformations could be found in embryos of all glyphosate treatments but with rates below 20% (data summarized in Table 3). All glyphosate treatments were significantly different from the control. Among the malformations recorded, lightly pigmented embryos and larvae were particularly frequent (Fig. 4C). Furthermore, reduced eye size occurred regularly (Fig. 4B), and some individuals suffered from cardiac or yolk sac oedemas (Fig. 4E). Two individuals showed a notable shortening of the tail (Fig. 4D). Deformations of the spine at 96 hpf were observed surprisingly rarely, despite the high rates of tail and spine malformations at 24 hpf.

Figure 4 Danio rerio larvae at 96 hpf.

(A) Control; (B) upper larvae with smaller eyes compared with control individual below; (C) right larvae with light pigmentation compared with control individual on the left; (D) larvae with a shortened tail; (E) larvae with cardiac oedema; (F) larvae with spine deformation; (D–F) larvae show smaller eyes as regular, additionally; as seen at three to five times magnification under a stereo microscope (Stemi 2000-C; Zeiss, Oberkochen, Germany).

In contrast to the glyphosate results, unbuffered 7dSh, thus at a neutral pH, did not induce mortality throughout any of the experiments, and no developmental delays or malformations occurred in 7dSh-exposed embryos. Sublethal effects were limited to a slightly decelerated heart rate (140 bpm), which corresponded to the 10 μM glyphosate treatment (138 bpm) but was eight bpm lower than in embryos of the negative control (ANOVA with Dunnett’s test, p < 0.001), and an accelerated hatching over time (Cox regression, p < 0.05) compared with the negative control. At 60 hpf, the hatching rates in the 7dSh-exposed group and the negative control were similar but clearly below those of the groups exposed to low glyphosate concentrations. At 72 hpf, 7dSh outpaced the negative control and converged on the results of 10 to 100 μM glyphosate treatments. In contrast, the hatching success of 7dSh-exposed embryos at 96 hpf did not differ from that of the negative control.

Glyphosate at pH 7

When the glyphosate solutions were adjusted to pH 7, almost no mortality or developmental delays occurred, and malformation rates were below 10% but were still significantly elevated in 1 and 10 mM treatments (likelihood ratio χ2, p < 0.001; Table 3). In the concentration range of 10 to 500 μM, heart rates showed a similar trend to those in unbuffered treatments but at a lower level: bpm decreased with increasing concentration (Fig. 2B). Still, treatments between 100 and 500 μM differed significantly from the negative control (Tukey’s HSD, p < 0.01). At 750 μM, heart rates increased again, with a higher frequency than at 250 and 500 μM. At the two highest concentrations (1, 10 mM), heart rates were, on the one hand, marginally decelerated (1 mM) and on the other hand, marginally accelerated (10 mM) compared with the negative control. Thus, it seems that there is a turning point between 500 and 750 μM, at which the relationship between increasing concentration and heart rate shifts from deceleration to acceleration in comparison with the negative control.

As already seen for lower concentrations in unbuffered treatments, glyphosate tends to induce early hatching, even at the lowest concentration and independently of concentration. This effect unfolded to its true extent in the pH-neutral treatments (Fig. 2D). At least twice as much larvae had hatched across all glyphosate treatments at 60 hpf compared with the negative control. After 72 hpf, all larvae were hatched in glyphosate treatments, except for single individuals that hatched at 96 hpf or did not hatch at all, whereas in the negative control, only 53% of the embryos were hatched at 72 hpf and even about 15% remained unhatched at 96 hpf.

pH range

In a first step, one mM glyphosate was tested at pH 3, 4, 5, 6, 7 and 8 in comparison with negative controls at the respective pH but without the pesticide (Fig. 5). Mortality was 100% for both treatments at pH 3, independent of the presence of glyphosate. Only a single individual survived the first 12 hpf. In contrast, only one individual died throughout all other exposures within 96 hpf. Morphological aberrations described for high glyphosate concentrations under unbuffered conditions (Figs. 1 and 3) also applied to low pH treatments, independent of glyphosate addition. Concerning sublethal endpoints, results between different acidities in the range of pH 4 to 8, as well as between control and glyphosate within the same pH range, were inconspicuous for the most part. Thus, the pH 3 to 8 series was tested just once, and subsequent testing concentrated on the range from pH 3 to 4. Thus, in the next step, pH 3, 3.25, 3.5, 3.75 and 4 were investigated in detail. As embryos exposed to pH 3.75 and 4 did not show any prominent effects, only a single run was conducted, and the final testing scheme was determined from pH 3 to 3.5 in 0.1 increments. Additionally, a test with unbuffered glyphosate at a one mM concentration (which resulted in a pH of 3.2 in the test solution) was included for direct comparison.

Figure 5 Mortality over time as percentages of embryos exposed to the pH control (A) and glyphosate (B).

Respective concentrations of glyphosate are given in brackets. Results from unbuffered treatments (50 μM–10 mM glyphosate; highlighted in red) are combined with pH range results and positioned according to their measured pH. Treatments not conducted in the pH control or glyphosate scheme are labelled n.a. (not available). Significant differences from the negative control are marked with asterisks (*), except glyphosate pH 3.4 with an additional significant comparison between unbuffered and pH 7 treatment. Significances between pH control and glyphosate treatments within respective pH ranges are denoted with letters (a and b) (Cox regression, p < 0.01).

Mortality decreased with increasing pH. Treatments with a pH of 3.2 and lower induced 100% mortality after 96 hpf. Whereas embryos exposed to pH 3 and 3.1 died within 48 hpf at the latest, embryos in pH 3.2 treatments survived considerably longer (Fig. 5). Apart from pH 3.5 without glyphosate, all treatments showed elevated mortality rates compared with the negative control (Cox regression, p < 0.05). There were no differences between control and glyphosate treatments with corresponding pH values, except for the elevated mortality in unbuffered glyphosate compared with the respective pH 3.2 control.

Compared with the negative control, hatching was significantly delayed and also reduced in both glyphosate and pH control treatments (Cox regression, p < 0.001). Whereas 30% of the control embryos hatched at 60 hpf, in the pH control and glyphosate exposures, the hatching rate at 60 hpf was consistently below 5% (see Supplementary File, hatching rate). The tendency toward glyphosate-induced premature hatching at 60 hpf that was observed in pH-neutral treatments was not evident at low pH. Although not statistically significant (except for pH 3.5: Cox regression, p < 0.001), embryos exposed to glyphosate tended to hatch earlier and more frequently than embryos in the respective pH controls (see also Table 3).

Heart rates were significantly lowered by glyphosate at pH 3.3 to 3.5, as well as by the corresponding control pH treatments (Steel-Dwass, p < 0.001). Differences between glyphosate and the respective controls at the same pH value could only be detected when the full pH range dataset (including results for pH 3 to 8) was analyzed. At a pH between 5.55 and 6.02, glyphosate elevated the embryonic heart rate significantly compared with pH controls (TableCurve 2D v5.01; Fig. 6). Developmental delays and malformations occurred in the low pH treatments, but they did not vary in a pH-dependent manner, and there was no detectable difference between glyphosate and the respective pH controls.

Figure 6 Mean deviation of the heart rate from the negative control in percent as a function dependent on pH; pH control: y = a + bx + cx1.5 + dx2.5 + ex0.5; glyphosate: y = a + bx + cx2 + dex + e (lnx)2.

Colored areas mark the 95% confidence interval. Green lines confine the pH range (between pH 5.55 and pH 6.05) in which the 95% confidence intervals of pH control and glyphosate do not overlap and thus, where heart rates differ significantly from each other.

Comparison

When datasets for the unbuffered glyphosate treatment and the pH range were merged regarding mortality in relation to pH (Fig. 5), interestingly, embryos exposed to unbuffered glyphosate showed higher mortalities at 500 and 750 μM compared with their 1 mM counterparts at pH 3.5 and 3.4, respectively. The unbuffered 750 μM treatment with a pH of 3.4, in particular, resulted in a mortality rate more than twice as high as that in the glyphosate pH 3.4 treatment (1 mM), mirroring mortality effects seen in treatments ranging rather between pH 3.25 and 3.3.

Discussion

The results we obtained for unbuffered glyphosate treatments are in close agreement with those reported in the literature. Fiorino et al. (2018) found mortality rates below 10% for concentrations between 0.005 and 10 mg/L (<100 μM, compare to Table 1), as well as 17% mortality in 50 mg/L (≈295 μM) treatments after 96 hpf. Uren Webster et al. (2014) and Stehr et al. (2009) found no effect on mortality in embryos exposed to concentrations up to 10 mg/L after 96 hpf. Those findings are in line with our results for unbuffered glyphosate treatments. By contrast, Bortagary et al. (2010) also tested higher glyphosate concentrations and detected mortalities of 35.4% (25 mg/L), 74% (75 mg/L) and 82% (150 mg/L) as early as 24 hpf. Their rates were remarkably higher than ours, in particular concerning concentrations of 25 and 75 mg/L; however, the exposure temperature in Bortagary’s set-up was 27.9 °C and thus about two K higher than in our experiments. Although zebrafish easily tolerate temperatures around 28 °C (López-Olmeda & Sánchez-Vázquez, 2011), a 2 K temperature difference may result in a higher metabolic rate. Thus, the effects of substances set in faster and might be more severe. Nevertheless, the abundance and characteristics of developmental delays and malformations from Bortagary’s experiment matched our findings. They described delays in separation of the yolk sac that very likely coincide with the lack of tail detachment observed in this study. Additionally, they found deformations of the spine and deformities in the tail area that are in line with the findings presented here. Similar effects occurred in glyphosate tests with Java medaka (Oryzias javanicus) embryos. Yusof, Ismail & Alias (2014) detected increased alterations in the curvature pattern at concentrations ranging from 100 to 500 mg/L glyphosate. Morphological alterations described for zebrafish embryos included yolk sac and pericardial oedema, deformations of the skeleton and spine and misshapen yolk sacs (Bortagary et al., 2010; Sulukan et al., 2017; Fiorino et al., 2018) and thus corresponded, at least partly, to our findings of oedema, light pigmentation and small eyes. In Java medaka, embryo malformations like disproportional head and body sizes, bends in tails and a lack of cornea were found (Yusof, Ismail & Alias, 2014). Similar findings were reported in Xenopus laevis embryos incubated with 1/5,000 dilutions of glyphosate-based herbicides. Those embryos showed alterations in cephalic and neural crest development and a shortening of the anterior-posterior axis, and when glyphosate was injected, optic vesicles were reduced, resulting in the formation of smaller eyes or in their complete reduction. In chick embryos, such an effect could be attributed to an increase in the activity of endogenous retinoic acid (Paganelli et al., 2010). Retinoic acid is a derivative of vitamin A that is crucial to various processes in vertebrates and thus ensures the proper development of zebrafish embryos on numerous levels (Holder & Hill, 1991; Marsh-Armstrong et al., 1994; Stafford & Prince, 2002; Kawakami et al., 2005; Keegan et al., 2005; Laue et al., 2008). As a consequence, interference in retinoic acid signaling has the potential to cause developmental delays and malformations during embryonic development in vertebrates. In this context, Sulukan et al. (2017) investigated potential correlations of malformation induction with carbonic anhydrases (CA), an enzyme family that is involved in various biological processes, including respiration, CO2 and ion transport, acid-base balance and the formation of reactive oxygen species (ROS). They found malformations in all treatments (1–100 mg/L), with simultaneous inhibition of CA and an increase in ROS leading to apoptosis, which was interpreted to be causally responsible for the observed malformations. As CA inhibition can directly or indirectly be related to increasing ROS (Sulukan et al., 2017) and, considering our results concerning pH, it might be questionable whether glyphosate itself is the sole inducer of the observed effects or whether a low pH contributes to the elevation of ROS levels and thus, might be at least as relevant regarding the development of malformations.

Although our results for mortality, developmental delays and malformations are in line with findings reported in literature, which did not consider glyphosate-induced pH decrease, they were not reproduced by glyphosate at pH 7. In a comprehensive analysis of data from pH control and glyphosate treatments, it can be seen that the described effects occurred independently of glyphosate or its concentration, as long as an acid milieu below pH 3.75 was provided. In the case of those endpoints in zebrafish embryos, sublethal effects may be detected only between pH 3.3 and pH 3.75. A pH lower than 3.25 induced 100% mortality within 96 hpf and within 12 hpf at pH 3, whereas pH 3.75 to pH 8 did not affect survival. However, it is not unlikely that sublethal effects caused by glyphosate itself are masked by the severe consequences of low pH.

Even though our data revealed that the effects detected here and also reported in the literature are likely to be caused mainly by low pH values, our data indicate that even at neutral pH, glyphosate affects embryonic development. Early hatching reported, for example, by Uren Webster et al. (2014) and Bortagary et al. (2010) at concentrations between 10 and 50 mg/L could be observed in pH neutral treatments in particular and was not concentration dependent. As all embryos seemed to be well developed at the point of hatching, early hatching should be interpreted as accelerated chorion degradation rather than as premature hatching. According to Kimmel et al. (1995), the hatching period of D. rerio starts as early as at 48 hpf already, and there is no developmental difference under normal conditions between larvae that hatch spontaneously and those remaining in the chorion. Thus, glyphosate at pH 7 seems to contribute to more spontaneous hatching instances. During embryo handling throughout the test, it was striking that the chorion of individuals within low pH and/or glyphosate treatments was more sensitive to physical contact, for example, while positioning embryos, compared with negative control embryos. Chorions were never injured during the test procedure but became dented easily. Through movements within eggs, embryos bulged out the dents by themselves rapidly. Hence, accelerated hatching might be correlated with a more damageable chorion that has ruptured sooner as a result of movements of the embryo with or without direct hatching intention. The fact that the effect of accelerated hatching was found to be more pronounced in pH 7 treatments might be due to the developmental status. Developmentally retarded embryos with late detachment of the tail in lower pH treatments were less agile than normally developed individuals in neutral treatments at the same time point and thus were less likely to rupture the chorion accidently. This assumption is basically supported by observations from Yusof, Ismail & Alias (2014), who found a thinning of the embryonic chorion of Java medaka following a 100 to 500 mg/L glyphosate exposure, and Zhang et al. (2017), who detected a decrease in surface tension of the chorion combined with an increase in locomotion that might lead to accelerated hatching in zebrafish embryos.

Regarding the heart rate, concentration dependency was still apparent at neutral pH, although it was less pronounced than in the unbuffered glyphosate treatments. At a high glyphosate concentration, the decreasing effect on the heart rate seemed to reverse, as the heart accelerated again. The outcomes of the pH range treatments corroborate the importance of pH to the heart rate but also the potential of glyphosate at one mM to accelerate the heart rate considerably, at least between pH 5.55 and 6.05.

Few studies have evaluated the effect of glyphosate on the heart rate in zebrafish embryos. Bortagary et al. (2010) reported a decrease in the heart rate following exposure to concentrations exceeding 25 mg/L, whereas Roy et al. (2016) studied the effect of glyphosate on the cardiovascular system in detail but solely regarding a single concentration. Their finding of a heart rate of 129 bpm in response to 50 mg/L glyphosate matches our results of 130 bpm at a slightly lower concentration of 250 μM glyphosate at neutral pH (compare Table 1), whereas it differed from the unbuffered treatment with a delta of about 4. Differences between studies might be attributed to temperature impact. Zebrafish embryos in this study were reared and observed at 26 °C, whereas Roy et al. (2016) conducted the entire test at 28.5 °C. The specific effect of temperature on the zebrafish heart rate has been addressed previously (Schweizer et al., 2017). The observed progressive decrease in heart rate might be correlated to structural abnormalities found by Roy et al. (2016) for the impact of glyphosate on heart development. In that study, the shape and size of the atrium and ventricle changed and decreased, the endocardium was thickened and the vascular network throughout the body was poorly connected. As a result, erythrocytes could not move properly and were slowed down on their way through the organism, contributing to the formation of cardiac oedema. Although there are no references for the effects of higher glyphosate concentrations on the zebrafish heart rate, our results are in agreement with a study on Java medaka embryos by Yusof, Ismail & Alias (2014). The authors used 100 to 500 mg/L glyphosate and detected only acceleration of the heart rate. A glyphosate concentration of 100 mg/L corresponds to a glyphosate concentration of 591.47 μM and thus relates approximately to the turning point between 500 and 750 μM, at which the decreasing trend in the heart rate was reversed.

Our study supports previous findings of negative effects of glyphosate on fish embryonic stages but, for the first time, distinguishes between direct consequences of the substance itself (hatching, heart rate) and indirect effects by lowering the pH of the surrounding medium (mortality, developmental delays, malformations). Since the primary target of glyphosate is absent in metazoans, the toxic side-effects of glyphosate may be caused either by secondary target sites or indirectly by inhibition of the shikimate pathway in the associated microbiome of the tested animals (Samsel & Seneff, 2013; Motta, Raymann & Moran, 2018) or impairment of symbiotic microorganisms in engineered plants with the glyphosate-insensitive EPSP synthase class II (Zobiole et al., 2010), respectively. Since the control compound 7dSh, which also inhibits the shikimate pathway, did not cause the side-effects observed for glyphosate, the observed effects were more likely caused by secondary target sites of glyphosate.

Although the tested concentrations were rather high compared with measured field concentrations, the inherent adverse potential of glyphosate in combination with further increasing application rates is worrying. Lethal and severe sublethal effects might be prominently driven by glyphosate’s ability to drastically lower pH but, given that its acidic character contributes largely to its efficacy, the herbicide’s action and the pH can hardly be considered separately. Due to their acidity, glyphosate anions have a high affinity to bind to cations and build neutral salt complexes, which enhance uptake through biomembranes (Rendal, Kusk & Trapp, 2011). Nevertheless, their high binding capacities also apply to strongly charged cations in soil (e.g., Fe3+, Al3+), whose binding inactivates glyphosate (Borggaard & Gimsing, 2008; Helander, Saloniemi & Saikkonen, 2012), which, in turn, seems to make leaching into surface water or groundwater—per se—a minor problem. However, glyphosate competes with phosphate for binding sites (Borggaard & Gimsing, 2008). In phosphate-saturated soils that prevail in areas of intensive agriculture, in which glyphosate is preferably applied, soil particles have a very limited capacity to bind glyphosate anions in addition to phosphate (Helander, Saloniemi & Saikkonen, 2012). Consequently, in these cases, leaching of glyphosate into waters must be taken into account. If glyphosate is leached into surface waters with little buffering capacity, side effects of the pesticide should not be underestimated (Folmar, Sanders & Julin, 1979; Howe et al., 2004; Relyea, 2005; Paganelli et al., 2010; Vera et al., 2010; Lopes et al., 2014; Uren Webster et al., 2014; Roy et al., 2016; Fiorino et al., 2018). In particular, when glyphosate comes in contact with organisms under slightly acidic conditions: Ehrl et al. (2018) showed that the permeability of glyphosate through biomembranes is increased in a slightly acidic compared to a neutral surrounding. Thus, it may be assumed that glyphosate effects increase with lower pH based on a facilitated uptake of glyphosate into cells.

Although recent studies even suggest, that the toxicity exerted by glyphosate in the field is rather caused by certain additives within applied formulations, including POEA, than by glyphosate itself (Tsui & Chu, 2003; Moore et al., 2012; Defarge, Spiroux De Vendômois & Séralini, 2018), we could clearly show that glyphosate as pure compound is able to induce adverse effects in a non-target organism. And in the light of potential health effects even in human beings induced by a chronic but low dose exposure to glyphosate (Samsel & Seneff, 2013; Swanson et al., 2014; Swanson, Hoy & Seneff, 2016), the ideal prospective solution for this issue would be a long-term replacement of glyphosate. The compound 7dSh, which bases its efficacy on interfering with the shikimate pathway, similar to glyphosate, seems to be a promising candidate. Its effectiveness in plants has been demonstrated (Brilisauer et al., 2019), and in contrast to glyphosate, it neither lowers pH nor induces severe effects in zebrafish embryos, as shown in our study. No mortality, developmental delays or malformations could be detected in embryos exposed to 1 mM 7dSh.

Conclusions

Although the severe effects detected seemed to be mainly caused by a low (glyphosate induced) pH, the compound glyphosate itself affects embryonic development in D. rerio on a sublethal level. Direct comparison between pH controls and one mM glyphosate at the respective pH showed only minor differences, but severe pH effects might mask the impacts of glyphosate. From a non-target perspective, it would be advantageous to buffer glyphosate to a neutral level to avoid the severest effects on fish if glyphosate were leached into the aquatic environment at higher concentrations. As glyphosate’s acidity elevates its efficacy as a herbicide, such a recommendation is highly unlikely to be implemented by producers and users. In view of the protection of non-target species from side effects, alternatives should be considered that might potentially substitute glyphosate in the long term, and we have shown that 7dSh may be a promising emerging candidate.

Supplemental Information

Supplemental Information 1 Raw data: glyphosate and pH treatments sorted by endpoints.

Click here for additional data file.

We thank Prof. S. Grond for providing facilities for the preparation of 7dSh.

Additional Information and Declarations

Competing Interests

Author Contributions

Data Availability

The authors declare that they have no competing interests.

Mona Schweizer conceived and designed the experiments, performed the experiments, analyzed the data, contributed reagents/materials/analysis tools, prepared figures and/or tables, authored or reviewed drafts of the paper, approved the final draft.

Klaus Brilisauer conceived and designed the experiments, performed the experiments, contributed reagents/materials/analysis tools, authored or reviewed drafts of the paper, approved the final draft.

Rita Triebskorn conceived and designed the experiments, contributed reagents/materials/analysis tools, approved the final draft.

Karl Forchhammer contributed reagents/materials/analysis tools, authored or reviewed drafts of the paper, approved the final draft.

Heinz-R. Köhler conceived and designed the experiments, contributed reagents/materials/analysis tools, authored or reviewed drafts of the paper, approved the final draft.

The following information was supplied regarding data availability:

The Supplemental File contains the raw data of the conducted experiments sorted by endpoints observed and treatments conducted.

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
