# Peer review of "How glyphosate and its associated acidity affect early development in zebrafish (Danio rerio)"

_PeerJ, doi:10.7717/peerj.7094_

## Round 0.1 · original submission · Minor Revisions

The reviewers were complimentary of your work and have only minor comments for clarification. Please address these comments and modify your manuscript accordingly.

·

Basic reporting

no comment

Experimental design

no comment

Validity of the findings

no comment

Additional comments

This is a well designed study that addresses the question of the degree to which acidification plays a role in glyphosate's toxicity to zebrafish. The results are sound and interesting, and the paper is definitely worthy of publication.

I only have a few minor comments.

ll 84-87:
"In the aquatic environment, where glyphosate is degraded to CO2 and aminomethylphosphonic acid (AMPA), which, in turn, is degraded to inorganic phosphate, ammonium and CO2, glyphosate’s half-life is estimated between 4.2 and 14 days (Giesy, Dobson & Solomon 2000, 87 Vera et al. 2010)."

It is not the case that it always degrades very quickly. Glyphosate can persist up to a year in seawater. See:

Mercurio P, Flores F, Mueller JF, Carter S, Negri AP. Glyphosate persistence in seawater. Mar Pollut Bull. 2014 Aug 30;85(2):385-90.

ll 240-241: "a lack of somite formation and eye development was not detected."

This is confusing. Somite formation was missing and eye development was not detected? Please rephrase for clarity.

l. 243
"Yet, even the embryos in glyphosate treatments that lacked tail detachment were able to move, albeit to a limited extent."
Is this a clear abnormality? Able to move prior to tail detachment?

ll. 282-283:
"around a third of the larvae did not hatch prior to 96 hpf and about 15% remained unhatched at that time point."
In other words, 33% remained unhatched at the previous time sample point?

Since the primary target of glyphosate is absent in metazoans, the toxic side-effects of

ll 437-440:
"Since the primary target of glyphosate is absent in metazoans, the toxic side-effects of glyphosate may be caused either by secondary target sites or indirectly by inhibition of the shikimate pathway in the associated microbiome of the tested animals (Samsel & Seneff 2013, Motta, Raymann & Moran 2018)."

You should reference this paper which shows that glyphosate has many off-target effects in Roundup-Ready soybean plants:

Luiz Henrique Saes Zobiole et al. Effects of glyphosate on symbiotic N2 fixation and nickel concentration in glyphosate-resistant soybeans. Appl Soil Ecol 2010; 44: 176-80.
These effects can not be due to the shikimate pathway since the plants have been engineered to be protected from EPSPS synthase inhibition.

Additional potential references:

A paper published in 2016 by Swanson et al. suggested that metabolic acidosis might be a significant contributor to glyphosate's toxicity to humans. It would be appropriate for you to reference this paper during the discussion section.

Swanson NL, Hoy, J, Seneff S. Evidence that glyphosate is a causative agent in chronic sub-clinical metabolic acidosis and mitochondrial dysfunction. Int J Hum Nutr Funct Med 2016; 4:32-52.


The recent paper below by Ehrl et al. shows that glyphosate crosses lipid membranes much more readily under acidic conditions compared to basic conditions. This would be a good reference for you to add as supporting evidence of increased toxicity under acid conditions.

Ehrl BN, Mogusu EO, Kim K, Hofstetter H, Pedersen JA, Elsner M. High Permeation Rates in Liposome Systems Explain Rapid Glyphosate Biodegradation Associated with Strong Isotope Fractionation. Environ Sci Technol 2018; 52(13): 7259-7268.
doi: 10.1021/acs.est.8b01004. Epub 2018 Jun 19.

Reviewer 2 ·

Basic reporting

"No comment"

Experimental design

"No comment"

Validity of the findings

"No comment"

Additional comments

The introduction is consistent with the proposal developed throughout the text; however, it is necessary to better describe the reason for the pH variations used in the study. It is important to include the possible consequences of pH variation at individual and population level and in which cases this happens in the environment.

It is also necessary to mention the sources of direct and indirect pollution by which pesticides come into contact with non-target organisms besides those included in the text, such as incorrect disposal of packaging and direct application (line 81). What is the pH variation that glyphosate causes in its solutions? Include reference (Line 103).

Include the reference for mortality of zebrafish with temperature variation (Line 344).

With regard to the discussions, it would be interesting to first describe the main findings in the research, and from this to align with the literature. It is also important to emphasize the importance of the parameters evaluated for the development of the embryos, when the fish suffer these changes demonstrated in the results, which can lead to adult life. In the environment, what are the consequences of animals with these deformations?

The idea of using the new herbicide is innovative. It is important to demonstrate that these new compounds do not affect biodiversity and it is possible to produce food organically with quality and quantity.

·

Basic reporting

The authors contribute to the understanding of glyphosate's influence on zebrafish, an important vertebrate species, at both neutral and low pH. They also compare glyphosate's action on the zebrafish to a novel (to science), natural substance (7dSh) that also interferes with the shikimate pathway, advancing our understanding of both glyphosate and 7dSh. I recommend this paper for publication.

I have the following suggestions.

Line 96. The authors state "it could be conceivable to develop ecologically sound alternatives that can potentially substitute for glyphosate." Because the authors are familiar with the work of Brilisauer et al. at Tübingen University, and worked with the cyanobacteria antimetabolite 7dSh in this study, showing that "it neither lowers pH nor induces severe effects in zebrafish embryos (Line 465), the authors might consider reworking Line 96. For example, "it could be conceivable to develop the cyanobacteria antimetabolite 7dSh to substitute for glyphosate." I suggest the authors either say this or acknowledge that there are already ecologically sound alternatives to glyphosate for weed control and that 7dSh might be an important addition to these technologies and methods (such as those practiced in organic agriculture). For example:

Dayan, Franck E., and Stephen O. Duke. “Natural Compounds as Next-Generation Herbicides.” Plant Physiology, vol. 166, no. 3, 2014, pp. 1090–1105. JSTOR, www.jstor.org/stable/43191531.

Line 66. Help the reader understand which list glyphosate tops (ie, sales or usage).

Line 70. The IARC's designation of glyphosate as a probable carcinogen is important to note, but consider adding citations that expand the reader's understanding of other possible adverse effects on human health from glyphosate exposure. Such as:

Swanson NL, Leu A, Abrahamson J, Wallet B. Genetically engineered crops, glyphosate and the deterioration of health in the United States of America. J. Organic Systems 2014; 9: 6-37.

Samsel, A., Seneff, S. Glyphosates suppression of cytochrome P450 enzymes and amino acid biosynthesis by the gut microbiome: pathways to modern diseases. Entropy 2013; 15(4): 1416-1463. (Which is referenced elsewhere in the paper.)

Line 78. Reference to microbial degradation of glyphosate is from 1977. Is there more current research available for citation?

Line 80. Perhaps here is a place to insert a sentence or two about the surfactants and adjuvants used with glyphosate, that have their own toxicity issues. Glyphosate is never used alone for agricultural applications. Adverse effects from spray drift and runoff are impacted by the addition of these chemical components in glyphosate formulations. Such as:

Tush, D., Loftin, K.A., and Meyer, M.T., 2013, Characterization of polyoxyethylene tallow amine surfactants in technical mixtures and glyphosate formulations using ultra-high performance liquid chromatography and triple quadrupole mass spectrometry: Journal of Chromatography A, v. 1319, p. 80-87, doi:10.1016/j.chroma.2013.10.032.

"We demonstrate that non-target effects below toxicity thresholds are not due to the declared active ingredient G but undoubtedly to the formulants alone and in formulations, as previously documented for some instances. Their endocrine-disrupting action can be explained by the membrane disruptions, where the steroidogenic enzyme aromatase is located in the endoplasmic reticulum, but may also be due to possible direct enzymatic interactions that could be synergistic with heavy metals (see below). The endocrine disruption by GBH has been observed in vivo by our group in several instances, on the androgen/estrogen balance that aromatase controls, after short-term Roundup treatment and after long-term exposure."(Tush et al. 2013)

The paper can stand alone without references to adjuvants in glyphosate formulations, but because scientific experiments and observations have shown membrane disruptions and other endocrine-disrupting action by commercial formulations, it may strengthen the impact of the authors' results that show that glyphosate alone can also cause sublethal adverse events.

Experimental design

No comment.

Validity of the findings

Results are thorough and informative, as is the data and conclusions.

Line 254. It would be helpful if it was more clearly stated that 7dSh did not induce mortality or malformations at any of the pH levels tested.

Line 369-375. Discussion of the elevated ROS is an important issue. Some discussion of elevated ROS disrupting enzymes and the importance of this would be welcome, but not necessary.

I look forward to seeing this paper published in PEERJ.

---

## Round 0.2 · accepted · Accept

Thank you for your efforts to revise your manuscript and address reviewer comments.